# Variability of Bacterial Homopolysaccharide Production and Properties during Food Processing

**DOI:** 10.3390/biology11020171

**Published:** 2022-01-21

**Authors:** Marion Nabot, Marie Guérin, Dharini Sivakumar, Fabienne Remize, Cyrielle Garcia

**Affiliations:** 1QualiSud, University of Montpellier, UMR QualiSud, 34398 Montpellier, France; marion.nabot@cirad.fr (M.N.); marie.guerin@univ-reunion.fr (M.G.); 2UMR QualiSud, Université de La Réunion, 7 Chemin de l’Irat, F-97410 Saint Pierre, 97410 Réunion, France; 3Department of Horticulture, Tshwane University of Technology, Pretoria 0001, South Africa; SivakumarD@tut.ac.za; 4SPO, Université de Montpellier, INRAE, Institut Agro Montpellier, 34000 Montpellier, France; fabienne.remize@inrae.fr

**Keywords:** exopolysaccharide, texture, rheology, natural polymers

## Abstract

**Simple Summary:**

Bacteria can produce diverse homopolysaccharides (HoPSs), which are polymers of glucose, fructose or galactose. The synthesis of these compounds is catalyzed by glycosyltransferases. The range of HoPS sizes is very large and mostly depends on the carbon source in the medium and the catalyzing enzyme; however, factors such as nitrogen nutrients, pH, water activity, temperature and duration of bacterial culture also impact size and yield. The sequence of the polymerization enzyme influences the structure of the HoPS, by modulating the type of linkage between monomers, both for the linear chain and for the ramifications. HoPSs’ size and structure influence its rheological properties. As a consequence, the choice of catalyzing enzyme and the modulation of environmental factors open ways to guide the production of specific HoPSs in foods by bacteria. This approach presents many advantages to propose applications of bacterial HoPS to texture foods, either by *in situ* or *ex situ* production, but requires a better knowledge of HoPS production conditions.

**Abstract:**

Various homopolysaccharides (HoPSs) can be produced by bacteria: α- and β-glucans, β-fructans and α-galactans, which are polymers of glucose, fructose and galactose, respectively. The synthesis of these compounds is catalyzed by glycosyltransferases (glycansucrases), which are able to transfer the monosaccharides in a specific substrate to the medium, which results in the growth of polysaccharide chains. The range of HoPS sizes is very large, from 10^4^ to 10^9^ Da, and mostly depends on the carbon source in the medium and the catalyzing enzyme. However, factors such as nitrogen nutrients, pH, water activity, temperature and duration of bacterial culture also impact the size and yield of production. The sequence of the enzyme influences the structure of the HoPS, by modulating the type of linkage between monomers, both for the linear chain and for the ramifications. HoPSs’ size and structure have an effect on rheological properties of some foods by their influence on viscosity index. As a consequence, the control of structural and environmental factors opens ways to guide the production of specific HoPS in foods by bacteria, either by *in situ* or *ex situ* production, but requires a better knowledge of HoPS production conditions.

## 1. Introduction

Homopolysaccharides (HoPSs) or homoglycans are polymeric compounds that have the particularity of being composed of a single type of monosaccharide. They include starch, cellulose, pullulan, glucan and inulin, which do not contain functional groups other than the hydroxyl group, contrarily to chitosan, alginate or carrageenans [1]. They can either be linear, such as amylose, or ramified, such as amylopectin. These compounds are produced by animals, plants, algae, fungi and bacteria. 

Bacterial HoPS can be water soluble and produced in foods, especially fermented foods, in which they contribute to the functional properties. They belong to the family of exopolysaccharides (EPSs), which means that they are produced extracellularly. As such, their production has been mainly studied from lactic acid bacteria (LAB), but others, such as *Bacillus*, acetic acid bacteria, *Streptomyces*, *Zymomonas*, *Pseudomonas* or *Paenibacillus*, can produce HoPSs [1]. One particularity of bacterial HoPSs is that they can be produced at a high level, up to 70 g L^−1^ when conditions are optimized in *Bacillus subtilis* Natto [2,3]. 

Many studies investigated EPSs’ bioactivites, and it is now recognized that EPSs exert anticancer, immunoregulatory (heteropolysaccharides), antidiabetic, antioxidant, anti-inflammatory, hypolipidemic and hypoglycemic effects (for a review see [1]). The observed effects change according to the producing strain and the subsequent nature of the produced EPS. As such, dextran obtained from the fermentation with *Weissella* spp. strains demonstrated antioxidant potential, just like heteropolysaccharides (HePS) from the same species [4,5], and antifungal activity but no antibacterial activity against standard bacterial strains, as opposed to mannan produced from *Weissella confusa,* for example [6,7]. In addition, as dietary soluble fibers, some are investigated for prebiotic effect [8]. Apart from bioactivity, HoPSs are produced for industrial applications: cosmetics, tissue engineering, packaging or edible coating, agriculture as biosurfactants and petroleum industry [9]. In the food industry, they play a role in product texture and water retention. They have been extensively studied in dairy and bakery products and to a minor extent in beverages [10,11,12,13]. The long duration of fermentation is a positive factor that allows bacteria to produce EPS [14].

Naturally occurring polymers are receiving renewed attention as highly biocompatible and environmentally friendly materials. The HePS from LAB have long been characterized since they play an important role in the rheology, texture and “mouthfeel” of fermented milks and were studied for some other fermented products. HoPS have long been evaluated to a lesser extent; however, recently, the most notable advances in LAB-EPS research have been related to HoPSs as they could be considered as a promising alternative to conventional additives given their higher level of synthesis [15].

From an applied point of view, an improvement of food functional properties can be expected from HoPS production through inoculation of selected strains and process optimization. Hence, this review aims to draw a clear picture of the possibilities of modulation of EPS levels in fermented foods from the variability of bacterial sources and through the adjustment of environmental conditions. 

## 2. Variability in Structure and Size

### 2.1. Different Families of Homopolysaccharides

HoPSs from bacteria are mainly formed from sucrose, by polymerization of either glucose or fructose. The enzymes involved belong to the hexosyltransferase family. The synthesis depends on the involved enzyme, which triggers the type of link between hexoses. 

Four types of bacterial HoPSs (Figure 1) are described and divided into subclasses based on the type of linkage and the position of the carbon involved in the linkage: α-glucans (dextran, alternan, reuteran and mutan), β-glucans, β-fructans (levan, inulin) and α-galactans [16]. These four groups contain polysaccharides with different types of linkage, molecular weights or lengths and levels of ramification. 

Generally, the molecular weights of HoPSs range from 4.0 × 10^4^ to 1.8 × 10^9^ Da. Some molecular weights are described for reuteran (2.8 × 10^7^ Da), levan (2.0 × 10^6^ Da) or inulin-like fructan (1.0 × 10^7^ Da) [17,18]. The size of dextrans varies between 10^4^ Da for *Acetobacter tropicalis* to 10^9^ Da for *Oenococcus kitaharae* [19,20]. The size of the EPS molecule may depend on the carbon source in the culture medium (Table 1). Polak-Berecka et al. [21] showed that the molecular weight of dextrans synthesized by *Lactobacillus rhamnosus* grown on five carbon sources, i.e., glucose, galactose, sucrose, maltose or lactose, were around 5.0 × 10^5^, 3.9 × 10^6^, 11.1 × 10^6^, 1.9 × 10^5^ and 7.0 × 10^6^ Da, respectively.

### 2.2. α-Glucans

The α-glucans are usually classified into four groups depending on the type of links and sequence in the HoPS: dextrans, alternans, reuterans and mutans.

Dextrans are mainly composed of a linear chain, e.g., 95% in *Leuconostoc mesenteroides* or fully linear for *A. tropicalis* [19,29]. Dextrans can be linked by α(1→6) glycosidic bonds and branched at the carbon atom at position 3 or, less frequently, at positions 2 and 4 [30]. Alternan contains in nearly equal proportion α(1→3) and α(1→6) linkages, which alternate on the chain with some degree of α(1→3) branching. 

Glucansucrases gather dextransucrase (EC 2.4.1.5) and alternansucrase (EC 2.4.1.140). The enzymes transfer a d-glucosyl residue from sucrose to a glucan chain and release d-fructose. Those enzymes belong to the glycosyl hydrolase (GH) 70 family and descriptions can be found on the CAZy (Carbohydrate-Active Enzyme; www.cazy.org, accessed on 25 October 2021) and BRENDA (The Comprehensive Enzyme Information System; www.brenda-enzymes.org, accessed on 25 October 2021) databases. Until 2016, GH70 enzymes were only described in lactic acid bacteria (LAB) [31].

Dextransucrase is largely described in *Leu. mesenteroides* and in others LAB such as *Latilactobacillus curvatus, Leuconostoc citreum, Leuconostoc lactis, Limosilactobacillus reuteri, Liquorilactobacillus hordei, O. kitaharae, Pediococcus pentosaceus, Weissella cibaria* and *W. confusa*. *A. tropicalis* and *Streptococcus mutans* also produce a dextransucrase [19]. It mainly catalyzes the formation of α(1→6) linkages, plus possibly side-chains linked with α(1→2), α(1→3) or α(1→4) linkages. The size and structure of dextrans are mainly affected by the sequence and structure of dextransucrase. Mutagenesis has shown dextransucrase regions involved in dextran size and other amino acids of the protein sequence implied in branching [32,33].

Alternansucrase is produced by *Leu. mesenteroides* and *Leu. citreum*. As for dextransucrase, the role of certain amino acids of the enzyme is of high importance regarding the size of the polysaccharide and the type of linkages. For instance, mutation of L940 increases the percentage of α(1→6) linkages and L940W leads to absence of linkages of the α(1→3) type and more oligo- than polysaccharides [34]. Similarly the nature of amino acids involved in acceptor sub-sites was shown to modify the linkage type [35]. 

Reuteransucrase, responsible for reuteran synthesis, differs from dextransucrase as it produces a HoPS containing mainly α(1→4) linkages, interconnected via α(1→6) bridges [17]. Slight differences in the sequence, as weak as a triple amino acid mutation, can lead to differences of HoPS structure from 70% α(1→4) linkages to 80% α(1→6) linkages [17]. Reuteransucrases are produced by *Lim. reuteri* [36], for instance, the reuteran from the probiotic bacterium *Lim. reuteri 121* contains α(1→4) and α(1→6) glycosidic linkages in proportions of 58% and 42%, respectively [36].

Mutans are polymers consisting of glucose residues containing more than 50% of α(1→3) glucoside linkages, mainly associated with α(1→6) linkages. They are produced by *Lim. reuteri, Leu. mesenteroides* and several strains of *Streptococcus* [30].

The properties of dextran, regarding rheology, depend on its structure and size. The shorter the linear chain, the lower the viscosity [19,20]. Other factors, such as pH, the presence of maltose or temperature, play a role in the structure and size of the produced HoPS dextran [37,38,39]. 

### 2.3. β-Glucans

Most β-glucans from bacterial origin are unbranched glucose polymers linked via β(1→3) glycosidic bonds. In some cases, trisaccharide repeats linked by β(1→3) form a linear chain, and ramifications are formed from β(1→2) glucose units. They are produced by *Levilactobacillus brevis*, *Pediococcus claussenii* [40], *Pediococcus parvulus*, *Oenococcus oeni* [41] and other *Pediococcus* or *Lactobacillus* species [42,43]. The enzyme is a 1,3-β-glucan synthase (EC 2.4.1.34), which uses intracellular UDP-glucose as a substrate.

In addition, acetic acid bacteria, such as *Acetobacter* and *Gluconobacter,* produce long-chains of β(1→4) glucan, called bacterial cellulose, and adopt a ribbon-like secondary structure [44].

Viscosity of β-glucans is directly related to their molecular weight, molecular structure, solubility in water and food matrix [45]. The impact of the linkage type on the litheness of the polysaccharidic chain is recognized: β(1→4) bonds result in stiffer chains compared to α(1→4) or β(1→3) bonds [46].

### 2.4. Fructans

Inulin is composed of branched β(2→1) fructose linkages whereas levan has β(2→6) or α(2→1) and α(2→6) fructose linkages. 

Bacterial inulin is obtained from inulosucrase (EC 2.4.1.9) activity. It catalyzes the transfer of a fructose residue from sucrose to form a polysaccharide chain connected by β(2→1) fructosyl linkages, resulting in the synthesis of high-molecular-weight inulin polymers. Inulosucrase is found in *Bacillus* sp., *Leu. citreum, Lactobacillus johnsonii, Lactobacillus gasseri, Lim. reuteri, Paenibacilllus macerans, W. confusa* and *W. cibaria* [27].

Levansucrase (EC 2.4.4.10) catalyzes the transfer of d-fructosyl residues from fructose to yield β(2→6) osidic bonds, which characterize levan. Many bacteria produce this enzyme, such as *Bacillus licheniformis*, *B. subtilis*, *Fructolactibacillus sanfranciensis*, *Gluconoacetobacter* sp., *Gluconobacter oxydans*, *L. gasseri*, *Leu. mesenteroides*, *Lim. reuteri*, *Paenibacillus polymyxa*, *Pseudomonas* spp. and *Zymomonas mobilis*. *L. reuteri 121* is known to produce a linear levan [47]. *Leu. mesenteroides* NRRL B-512F produces levansucrase in addition to dextransucrase. Levansucrase is responsible for at least 25% of the reducing sugars released when grown in the presence of sucrose. This corresponds to a significant level of glucose and explains why fructose repression of dextransucrase is observed in *Leu. mesenteroides* [48].

### 2.5. α-Galactans

Galactans are a relatively less abundant class of polysaccharides, and their main structure is characterized by a chain of galactose units linked by α(1→6) and α(1→3) bridges [49]. *W. confusa* KR780676 produces a linear galactan containing α-(1→6)-linked galactose units [28].

## 3. Gene Regulation and Variability of HoPS Production Level

The bacterial genes responsible for EPS production and the promotion of various secondary functional features to the bacteria have been explored. Studies helped to specifically identify the genes that are crucial in the production process of bacterial HoPSs. Depending on their great structural diversity, EPSs can be produced by microorganisms via different pathways. HoPSs are unique in that they are synthesized by extracellular glycan-saccharides using mostly sucrose as the glycosyl donor (fructose or glucose). Described in the genera *Weissella, Leuconostoc, Lactobacillus* and *Pediococcus,* the synthesis pathway of HoPSs generally occurs extracellularly [50]. HoPSs are produced by transglycosylases (glycansucrases), which are able to use the energy of the osidic bond of sucrose to catalyze the transfer of a corresponding glycosylated fraction. The transfer of monosaccharide in a specific substrate (e.g., sucrose) to the medium results in the growth of polysaccharide chains [51]. Most of the HoPS producing-LAB harbor only one glucansucrase gene; however, some LAB genomes exhibit more than one gene encoding hexosyltransferases and are, thus, able to synthesize different HoPSs. For instance, the *Leu. mesenteroides* NRRL B-512F strain coding both a glucansucrase and a fructansucrase can produce levan in addition to the dextran usually produced [52].

The subsequent production of HoPSs by LAB can be regulated by the production level and the activity of these enzymes. Of note, the production of dextransucrase is dependent on the presence of sucrose for all species, except some *Streptococcus* spp. On the contrary, reuteransucrase is constitutively expressed whatever the sugar [53]. The optimum sucrose level varies depending on the strain, generally between 2% and 5% for dextransucrase but up to 10% for reuteransucrase. On the other hand, pH of the medium does not influence the level of production of the enzyme but could act on its stability [37]. Divalent cations, Ca^2+^, Mg^2+^ and Mn^2+^ and dextrans activate and stabilize the enzyme of *Leu. mesenteroides*.

The production of EPSs can be influenced by changing the nature or quantity of substrates or available nutrients as well as by changing the pH, water activity, temperature and oxygen concentration of the culture medium [36]. The structure of the EPSs and their content vary according to the sources of carbon, nitrogen, phosphorus or sulfur. Sucrose is the most commonly used substrate for the synthesis of HoPSs and often the best substrate for HoPS production yield. LAB are sensitive to pH and the nature and amount of sugars. These factors must be taken into consideration for the production of EPSs in fruit and vegetable products with respect to their composition [52]. The specificity of carbon substrate differs from species to species. The temperature and yield depend on the different strains used (Table 2). 

Regulation at constant pH favors better HoPS yields, because glycohydrolases are activated when acidification occurs due to lactate production, around pH 5. When the maximum concentration of lactate is reached in *L. rhamnosus* culture, the polysaccharide yields decrease due to enzymatic digestion. It usually corresponds to 24 to 48 h of fermentation [21]. The study by Polak-Berecka et al. [26] showed that culture conditions have a strong effect on the production of EPSs by *L. rhamnosus* and scaling from flask to fermenter can increase EPS biosynthesis by 175.8% in commercial production processes. 

The viscosity of aqueous solutions is partly linked to HoPS level, molecular weight, degree of chain stiffness, radius of gyration of the molecule and the presence of ionizable groups that confer polyelectrolytic behavior to the polymer [54]. Looijesteijn et al. [55] reported that the main factor in the viscosity intensifying ability of EPS was the molecular weight. In this study, *Lactococcus lactis* subsp. *cremoris* strains NIZO B40 and NIZO B891 produced EPS with lower molecular weight in a medium with a low glucose content. The chemical composition of these polymers remained unchanged, but the rheological properties of the solution changed.

## 4. Effect of Food Composition and Processing 

In the case of EPS synthesis occurring during food processing, the production yield will be influenced by the food environment on top of the experimental conditions of fermentation (Figure 2). 

### 4.1. Role of the Matrix Composition

The importance of the choice of the substrate was shown for HoPS production in cereals, influencing both the level of production or the structure of the HoPS [57]. Thereby, during sourdough production based on four different flours (buckwheat, quinoa, sorghum and teff) with a dextran-producing *W. cibaria* strain, EPS and oligosaccharide concentrations were observed to be inversely related. The highest dextran content was formed in sourdoughs prepared with buckwheat (4.2 g EPS kg^−1^ sourdough) and quinoa (3.2 g EPS kg^−1^ sourdough). Furthermore, the level of maltose in flour was found to be a key point for high-molecular-weight EPS synthesis. Maltose was found to be higher in wheat and lowest in buckwheat, resulting in the lowest levels of high-MW dextran in wheat sourdoughs [57,58]. 

Factors such as monosaccharide composition, type of linkage, side chains, molecular weight and their interaction with other constituents (mainly casein and ions) also affect the rheological functions of EPS lactobacilli in fermented milk products [59]. 

### 4.2. Addition of Ingredients

It is known that sucrose is first used by LAB for growth during the exponential phase and then can be used for dextran production during the stationary phase by EPS-producing strains liberating fructose. Sucrose addition was tested on fava bean flour fermented with *Leuconostoc* spp. and *Weissella* spp. to evaluate it potential in EPS production [60]. It was observed that the addition of sucrose strongly induced dextran and glucan production when compared to control fava bean dough fermented with the same starter. Glucan contents were found to be higher than dextran contents in all cases, with 1.86% to 3.67% and 2.57% to 4.33% in sucrose-enriched doughs versus 0.32% to 0.82% and 0.11% to 0.74% in control dough, respectively, as low- and high-branched dextrans were both included in glucans. Interestingly, the higher production of glucan noticed for *Leuconostoc* spp. compared to *Weissella* spp. in sucrose-enriched doughs could be related to the low-branched dextran preferentially produced by *Weissella* spp. while *Leuconostoc* spp. tend to produce more than one type of dextran.

Sucrose addition was also tested on legume-based sourdough as a potential approach to improve the use of legume flour in bakery products. The addition of 2% of sucrose during fermentation at a temperature of 30 °C allowed the production of linear dextran with 2.6% α(1→3) linked branches by a *W. confusa* strain isolated from spontaneous fermentation and promoted the technological properties of legume-based sourdough [61]. 

Strategies to promote sugar availability in tubers also include enzyme activity, e.g., inulinase was added in Jerusalem artichoke for simultaneous saccharification and fermentation with *Bacillus velezensis* LT-2. The maximum EPS yield (9.25 g L^−1^), benefiting from a shortened fermentation period by 26.67% and a significantly associated reduced cost, was obtained with the inulinase dosage of 18 U g^−1^ sugar. This condition provided sufficient monosaccharides while exhibiting no substrate inhibition and affected the bacterial genes’ regulation in the following aspects: impaired fructose inhibition, accelerated the synthesis of EPS nucleotide-sugars precursors, induced the transcription of the EPS synthetic gene cluster and strengthened the electron transport respiratory chain and transporter system [62]. 

The influence of fat content on EPS production was tested in meat fermented products with the aim of producing fat-reduced sausages. For a final fat content ranging between 17% and 33%, the EPS production observed in spreadable raw sausages ranged between 0.46 and 1.03 g kg^−1^ for the HoPS-producing strains *Latilactobacillus sakei* 1.411 and *Lat. curvatus,* 1.1928 resulting in reduced hardness and higher softness. However, no significant differences of HoPS production levels were noted for the different fat contents [63].

### 4.3. Pretreatments

Some pretreatments are considered to modulate the level of EPS production in the food matrix. This is the case with sprouting, which was recently tested for lentil flour [64]. The dextran levels synthesized *in situ* by a *W. confusa* strain were found to be slightly higher in sprouted lentil sourdoughs than in those with unsprouted lentils, with 9.7% and 9.2% w/w flour weight, respectively. 

It appears that other pretreatments influence the EPS production yield as is the case for thermal treatment. The heat intensity applied upstream of fermentation seems to be a determining factor, as demonstrated in a lupin-based milk alternative, where a more intensive heat treatment (ultra-high temperature) resulted in higher amounts of HePSs and better rheological and textural properties than pasteurization [65].

Nonthermal technology such as ultrasound was also used to shorten fermented products’ fermentation time. This technology allows to shorten the adaptation phase of microorganisms in the growth curve and enhance the membrane permeability, therefore allowing a mass transfer. This technic showed a tendency to improve the EPS production rate in kefir [66].

### 4.4. Conditions of Fermentation 

The role of fermentation conditions on *in situ* EPS production levels was shown on sucuk sausage, the most consumed fermented meat product in Turkey [67]. A similar tendency for the EPS production levels with both *Latilactobacillus plantarum* and *Leu. mesenteroides* strains was observed when fermentation temperatures of 14 to 18 °C and durations of 8 to 16 days were compared. The EPS production increased significantly with the prolongation of ripening period and fermentation temperature, from 4.68 to 18.96 mg EPS kg^−1^ dry matter, in a conversely proportional way to pH. This can be related with the temperature requirement and growth-associated dependency of EPS production depending on the mesophilic or thermophilic nature of the strain.

### 4.5. Traditional (Spontaneous) vs. Inoculated

Lactic acid bacteria EPSs from spontaneously fermented wheat bran sourdough have been studied under the prism of the production by dominant isolates [68,69]. However, the diversity or the level of EPS production in spontaneous fermented food in comparison with inoculated fermented food has not been characterized, and the question remains of interest for functional food optimization.

## 5. Technological Impact of EPSs in Foods, Depending on Matrix and Processing Conditions

The combination of the endogenous factors of the matrix and the exogenous factors relative to the applied processes greatly influences the synthesis and the properties of EPSs in foods and, ultimately, the quality of the final product (Table 3).

### 5.1. Applications of EPSs 

There are many potential industrial applications for LAB-originated EPSs based on their role as gelling, thickening, emulsifying, stabilizing, water-binding and viscosifying agents. EPS technological features are traditionally useful in fermented dairy foods manufacture such as yogurt, milk-based desserts, kefir, cultured cream or cheese. They can act as thickeners and texturizers conferring advantageous properties during the process. Their structure allows them to bind water to stabilize and increase the viscosity of the final product, to interact with milk proteins and micelles to improve the firmness of the casein network and at the same time to decrease syneresis [74].

HePSs, composed of several carbohydrate moieties, play an important role in dairy products as they can be produced from various sugars and notably lactose in the absence of sucrose [75,76]. It was shown that HePSs produced by utilizing different carbon sources vary only in the fraction of different sugars, while the composition remained the same, indicating that sugar type has no effect on the composition of HePSs despite its capacity to process the sugars into various monomer units [77].

HoPSs on the other hand, consisting mainly of glucose or fructose, are effectively used in cereal- and legume-flour-based products as a natural alternative to commercial hydrocolloids for the enhancement of both gluten-containing and gluten-free cereal- and legume-based products [71]. The exploitation of HoPS production is also particularly interesting in yogurt-like beverages or cheese alternatives aiming to mimic fermented dairy products, because the polysaccharides formed improve the textural and sensory properties and extend the shelf life of these plant-based products [74].

### 5.2. In Situ Production versus Ex Situ Addition 

LAB-derived EPSs can be added as additives in foods or formed *in situ*. EPSs can be considered for addition into food products during processing as a biopolymer, as an alternative that is easier to control than *in situ* production by LAB culture. The addition of adapted amounts of polysaccharide at a specific time-point during food processing can lead to the desired functional and physical characteristics of the end product. This use of dextran is currently allowed by the European Commission at levels of up to 5% in bakery products [78] provided it is listed as an additive in the ingredients on packaging. For instance, the *ex situ* production and the use of bacterial EPSs as ingredients enables wider application of this category of EPSs since, in this form, the polymer can be added to the food in greater quantities than it would be if synthetized *in situ* [79]. 

Regarding the *ex situ* addition, hydrocolloids such as xanthan or hydroxypropylmethylcellulose are commonly used in gluten-free formulations in order to improve the viscoelastic properties of doughs. Indeed, gluten-free products are characterized by low water absorption, changes in crumb characteristics, decreased bread volume and poor stability. The efficiency of *in situ* HoPS production on the structure and quality of gluten-free bread enhancement has been studied by comparing buckwheat and rice flour mix and supplemented with different *Lactobacillus* EPSs (levan and dextran) or hydroxypropylmethylcellulose by Rühmkorf et al. [71]. In this study, all supplements increased the specific volume and reduced the crumb hardness noticeably. However, the improvement in moisture content, baking loss and crumb hardness were higher with dextran produced by the *L. curvatus* TMW 1.624 strain. This specific dextran was found to have a higher water retention capacity and molar mass than other analyzed EPSs. A structure–function relation was suggested in which high weight, average molar mass and branching at position 3 of the glucose monomer foster a compact conformation of the molecule and higher water binding capacity, conferring a high efficiency to this *in situ* HoPS. It goes in the direction of more efficiency for EPSs formed *in situ* compared to that of those produced *ex situ* and then added externally, as previously reported for bread [80]. In addition, the *in situ* production of EPS is generally considered as a promising alternative to conventional additives [75]. *In situ* production of EPS by EPS-producing strains present many advantages since it has less impact on production cost than addition, isolation and purification of EPS for use as an ingredient, and it fits the consumer demand for products with fewer additives and “clean labeled” products.

### 5.3. Texture

Independently of their origin, EPSs have been classified based on their functionality. It has been established that the intrinsic properties of EPSs (e.g., composition, branching, charge, molar mass) drive inter-macromolecular associations through hydrogen bonds, electrostatic or ionic forces. These factors are a key point for protein interaction and complexations with bacterial cells that are of great importance in rheological properties of fermented products, pro- and prebiotics interactions and biofilm formation [81]. Based on this statement, a web-based platform-independent database of bacterial exopolysaccharides provides access to detailed structural. Taxonomic, growth conditions, functional properties and genetic and bibliographic information for EPS (http://www.epsdatabase.com) (accessed on 21 December 2021) in order to provide a fast access to functional EPS properties.

#### 5.3.1. Rheological Properties

The comparison of bread recipes using sourdough obtained from different fermented flours (buckwheat, quinoa, sorghum and teff) with a dextran-producing *W.*
*cibaria* strain has shown an impact of the flour matrix on the bread’s rheological properties. For instance, the crumb hardness was reduced in buckwheat (−122%), teff and quinoa breads whereas a significant reduction of the staling rate was observed in buckwheat and teff breads in relation with the variability of HoPS production [57].

#### 5.3.2. Viscosity

Specific spatial properties of the macromolecular conformation of EPSs are reflected in the intrinsic viscosity value [η]. This hydrodynamic parameter is influenced by the chain length, the backbone stiffness, the charge and the molecular weight and can be considered a key parameter describing the thickening capacity of a polysaccharide. Intrinsic viscosity depends on polymer size and conformation as well as on polymer–solvent interactions. As [η] describes the volume occupied by a single polymer chain, a high [η] usually represents large molecules and stiff chains, whereas low [η] indicates small molecules and flexible chains [82]. In solution, HoPSs have generally low [η]. As such, levans have a very low [η] (>0.5 dL g^−1^) even at high molecular weights, and this can be related to their compact and spherical molecular structure [83]. Less compact HoPSs such as dextran have an [η] between 0.1 and 1.5 dL g^−1^ [84,85]. By comparison, polyionic EPSs such as xanthan have a more extended structure due to their charge repulsion and a greater [η] (up to 76 dL g^−1^) as ordered secondary structure can also affects [η] [86]. 

The impact of *in situ* produced EPS on acidification, rheology and texture was evaluated in legume protein-rich foods by comparing two strains of *Leuconostoc pseudomesenteroides* and *W. confusa.* Along with a higher amount of dextran, a clear improvement in rheological and textural parameters was observed in 5% sucrose-added fava bean pastes after fermentation [72]. Higher viscosity index and viscoelasticity values (firmness, consistency, cohesiveness) were observed, conferring an elastic-paste structure due to the presence of dextran that could interact with proteins and further strengthen the gel structure, compared to pastes fermented with the same starter having a liquid structure. The benefit of *in situ* dextran production in fava bean protein is emphasized, since the observed properties could not be mimicked by simply mixing dextran, organic acids and the protein concentrate.

Several studies following the effect of sucrose supplementation in sterile skim milk textural properties demonstrated that milk solidification depended on the concentration of sucrose added with EPSs produced by *Weissella* sp. TN610, *W. hellenica* SKkimchi3 and *Leu. pseudomesenteroides* PC strain from pickled Chinese cabbage [2,87,88]. In the case of EPS from *Leu. pseudomesenteroides*, good pseudoplastic rheological properties and an increased viscosity according to the EPS concentration were observed. It was observed that the viscosity was influenced by process parameters such as temperature and pH. Indeed, at the same shear rate, viscosity decreased with increasing temperature maybe in relation to an increase in molecular flexibility and loosened polymer structure at higher temperatures [88]. It was also observed that the viscosity of EPS decreased as pH increased from 3.0 to 9.0, in line with previous studies on *Sporidiobolus pararoseus* EPSs [89]. 

#### 5.3.3. Syneresis

The application of EPSs in the dairy industry is of high interest, especially for beverages and fermented milk products with the aim of reducing the amount of added milk solids. EPSs are able to texturize these products by improving viscosity without affecting aroma or taste and by reducing syneresis occurring along the fermentation process. However, these properties have only been connected to HePSs so far since they are characterized by high intrinsic viscosities linked to their β(1→4) linkages, leading to the formation of EPS solutions with high consistency, and by branching degree, which is directly related to the polymer stiffness [90].

## 6. Conclusions

The properties of HoPSs in food products open new opportunities for functional food development through their textural effects. As such, their use in plant-based foods aiming to mimic fermented dairy products or in cereal and legume flours as a natural alternative to additives, arouses growing interest. While they are composed of a single type of monosaccharide, the diversity of structures observed for these molecules is large, with a manifold combination of basic units and linkages leading to multiple linear or ramified chains, and various molecular weights. These factors on top of the yield of production strongly influence the HoPS characteristics.

A majority of the data concerns the culture media, and further investigations will be required before being able to predict the *in situ* HoPS production level, structure and functional effects according to the food matrixes and the sequence of the gene encoding glycosyl hydrolase. Nevertheless, production in foods can be controlled by the bacterial strain and the optimization of incubation parameters for bacterial growth and glycosyl hydrolase enzymes’ stability. Sugar availability as a substrate and enzyme activity are crucial for HoPS production in order to control the composition and environment of the food and to optimize the production.

## Figures and Tables

**Figure 1 biology-11-00171-f001:**
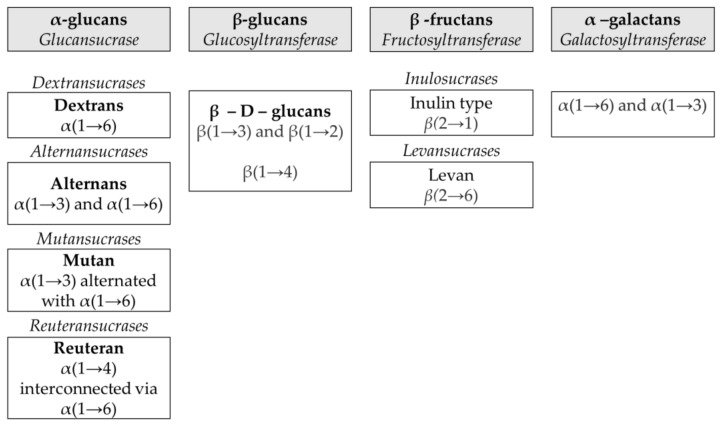
Classification of bacterial homopolysaccharides according to the enzymes involved and their linear chain linkages.

**Figure 2 biology-11-00171-f002:**
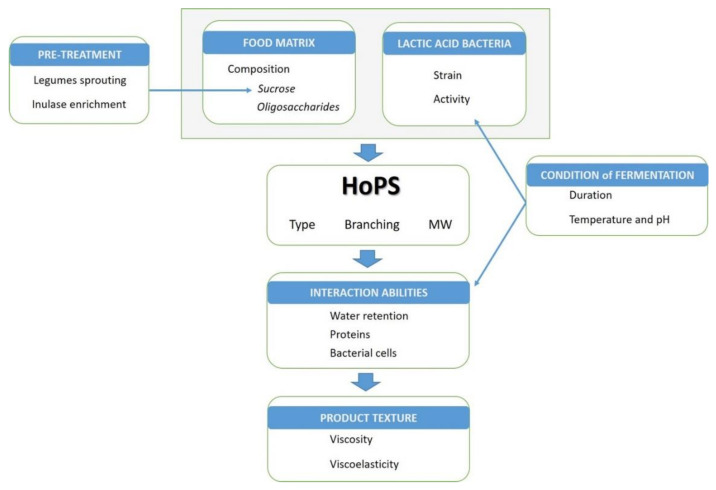
Relation between food matrix, processing parameters and product technological parameters.

**Table 1 biology-11-00171-t001:** Size and linkages according to the producing strain. HoPS: homopolysaccharide; Glc: glucose units; Fru: fructose units.

Strain	HoPS	Molecular Weight Range (Da)	Linkage	Reference
*Leuconostoc mesenteroides* TDS2-19	Dextran	9 × 10^7^	α(1→6) Glc linear	[22]
*Leu. mesenteroides* RTF10	Dextran	4 × 10^8^	α(1→6) Glc and branched	[23]
*Latilactobacillus sakei* MN1	Dextran	2 × 10^8^	α(1→6) Glc and partially branched in the *O*-3 position by a single α-glucopyranose unit (between 8.5% and 10.3%)	[24]
*Leu. mesenteroides* NRRL	Dextran	6 × 10^5^	95% α(1→6) Glc and 5% α(1→3) Glc	[16]
*Acetobacter tropicalis*	Dextran	1 × 10^4^ to 4 × 10^4^	α(1→6) Glc linear	[19]
*Limosilactobacillus reuteri* ML1	Reuteran		α(1→6) Glc and 4,6-disubstituted α-glucosyl units at the branching points	[17]
*Levilactobacillus**brevis* TMW 1.2112	β-Glucan		β(1 → 3) Glc ramified with β-Glc residues at position *O*2	[25]
*Lim. reuteri* 121	Levan	2 × 10^5^ to 2 × 10^6^	98% β(2→6) Fru and 2% β(1→2) and β(2→6) Fru branched	[18]
*Bacillus subtilis* Natto	Levan		β(2→6) fructofuranoside	[3]
*Lactobacillus rhamnosus*	Inulin	11 × 10^6^	β(2→1) Fru glycosidic and Fru branched at the β(2→6) position	[26]
*Lactobacillus gasseri* DSM 20604	Inulin	6 × 10^6^	β(2→1) Fru glycosidic	[27]
*Weissella confusa* KR780676	Galactan		α(1 → 6) galactose	[28]

**Table 2 biology-11-00171-t002:** Influence of culture conditions and strains on HoPS production yield.

Strain	MW (Da)	Optimum Temperature (°C)	pH	Medium Composition	Yield (g L^−1^)	Application	Reference
*Leu. mesenteroides* TDS2-19	8.8 × 10^7^	25	6.8	MRS ^1^ medium	71.23	Used in food industries as an emulsifier or as part of starter culture	[22]
*Leu. mesenteroides* RTF10	4.4 × 10^8^	30	4.8	CDM ^2^ 0.8% sucrose	1.25	Used as adjuvant and stabilizer in food industries	[23]
*Lat. sakei* MN1	1.7 × 10^8^	30		MRS 2% sucrose	1.72	Used for biofilm formation	[24]
*W. cibaria* MG1	7.2 × 10^8^	30	4.0–4.1	MRS 10% sucrose		Used in bakery for sorghum bread	[7]
*Lim. reuteri* ML1		35	4.7	Sucrose (purified enzyme)	5.12		[17]
*Lim. reuteri 121*		37	4.8	MRS medium sucrose	5.2		[56]
*Lim. reuteri* VIP	1.0 × 10^7^	37	3.6	MRS 10% sucrose		Used in bakery for sorghum bread	[7]
*B. subtilis* Natto		37	5.6–5.8	SM ^3^ sucrose 20%	70.60	Used in pharmaceutical industries as a commercial spore, on alginate matrix for repeated production of levan	[3]
*Weissella hellenica* SKkimchi3	20.3 × 10^4^	20	5	MRS sucrose 30%	74.00		[2]
*Lim. reuteri 121*	2.0 × 10^6^	37	4.5–5.5	In vitro enzyme			[18]
*L. rhamnosus*	11.1 × 10^6^	37	5.0	YNB ^4^	2.10		[26]
*Lim. reuteri* Y2	8.9 × 10^6^	37	3.7–3.8	MRS 2.5% sucrose or raffinose			[7]
*L. gasseri* DSM 20604	5.8 × 10^6^	35	5.5	sucrose	53.00		[27]

^1^ MRS is composed of sucrose, yeast extract, beef extract, anhydrous sodium acetate, ammonium citrate; ^2^ CDM, chemically defined medium; ^3^ SM, Spizizen medium; ^4^ YNB, yeast nitrogen base.

**Table 3 biology-11-00171-t003:** Influence of some food pretreatment and processing condition on EPS production and properties.

EPS	Strain	Processing Conditions	Effects	Reference
Dextran	*W. cibaria* MG1	Different fermented flours (buckwheat, quinoa, sorghum and teff) used as the basis for bread recipes using sourdough	Yield depended on the substrate and was highest in buckwheat and quinoa sourdough. The level of maltose in flour influences the MW of synthetized EPSs in sourdoughs (lower maltose in buckwheat resulting in the most high-MW dextran). Bread rheological properties were influenced by the flour matrix in relation to the variability of HoPS: reduction of crumb hardness in buckwheat (−122%), teff and quinoa breads; reduction of the staling rate in buckwheat and teff breads.	[57]
Dextran	*W. confusa* VTT E-90392	*In situ* EPS production in wheat sourdoughs, 10% enriched with sucrose or unenriched. Sourdoughs were used in baking at 43% of the dough weight.	*W. confusa* efficiently produced polymeric dextran (11–16 g/kg DW) from the added sucrose in wheat sourdough without strong acid production. The produced dextran significantly increased the viscosity of the sourdoughs. Application of dextran-enriched sourdoughs in bread baking provided mildly acidic wheat bread with improved volume (up to 10%) and crumb softness (25–40%) during 6 days of storage.	[70]
Dextran	*W. confusa* SLA4	Lentil flour sprouting	Increase slightly the dextran synthesis in comparison to nonsprouted lentil sourdoughs (9.7% and 9.2% w/w flour weight, respectively).	[64]
EPS	Kefir starter culture	Ultrasonic sound waves	The treatment allows kefir production in a shorter time by affecting the growth rate and lactic acid and EPS production rate.	[66]
EPS	*Latilactobacillus plantarum* 162R and *Leu. mesenteroides* N6	Ripening period and fermentation temperature	Increase of EPS production level, associated with hardness reduction of the fat-reduced products and lower loss and storage moduli, when the ripening period was prolonged and the fermentation temperature was higher.	[67]
Dextran Fructan, glucan Dextran	*Latilactobacillus curvatus* TMW 1.624 *Ligiactobacillus animalis* TMW 1.971 *Lim. reuteri* TMW 1.106	*In situ* production of various EPSs compared to the addition of hydroxypropyl-methylcellulose (HPMC)	Only HPMC and the dextran of *Lat. curvatus* TMW 1.624 retained water. The moisture content, baking loss and crumb firmness were improved the most by dextran of *Lat. curvatus* TMW 1.624. Structure analysis revealed that this dextran had the highest molecular weight of the analyzed EPSs (118–242 MDa) and was branched in position 3 (8–9%). A structure–function relation was suggested in which high weight, average molar mass and branching at position 3 of the glucose monomer foster a compact conformation of the molecule, enabling an increased water-binding capacity and promoting superior structural effects in gluten-free breads.	[71]
Dextran	*Leuconostoc pseudomesenteroides* DSM 20193 and *W. confusa* E3403	*In situ* production in legume protein-rich foods (fava bean protein concentrate)	Stabilization, prevention of protein aggregation. Improvement in rheological and textural parameters in sucrose-added pastes after fermentation. *W. confusa* exhibited a higher viscosity index and a more rigid character of formed gel values than *Leu. pseudomesenteroides* (1441 and 766 g sof viscosity index, 0.39 and 0.23 of relative viscoelasticity index tan δ, respectively).	[72]
Dextran	*W. confusa* DSM 20194, compared to probiotic strains (*Lat. plantarum* T6B10, *L. rhamnosus* SP1)	Quinoa flour subjected to desaponification and gelatinization prior to fermentation	The content of 35%, w/w of quinoa flour in water was determined as optimal regarding the viscosity parameter. The viscosity and water holding capacity increased during fermentation with *W. confusa*, as the consequence of the EPS synthesized, contrary to what observed for other strains (0.7 Pa s and 98% observed for *W. confusa* versus 0.2 Pa s and 69–70% for the other tested strains, respectively).	[73]

## Data Availability

No new data were created or analyzed in this study.

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
