# Peer review of "Variability of Bacterial Homopolysaccharide Production and Properties during Food Processing"

_biology, 2022, doi:10.3390/biology11020171_

Round 1

Reviewer 1 Report

This article discuss about variability of bacterial homopolysaccharide production and  properties during food processing. Manuscript needs following changes.

Comments:

Line 40: One particularity of bacterial HoPS is that they can be produced at high level…not all the microbes able to produce that hight amount of EPS. Mention microbial examples able to produce high content of EPS.

Line 42: Many studies investigated EPS bioactivites, and it is now recognized…is mentioned activities shown by all EPS? You can discuss specific examples of EPS and their properties.

In introduction part need to provide difference in hompolysaccharides in relation to composition, bioactivity, synthesis methods etc. What are the special properties of HoPS that make them different from other EPS. What is the importance of this article?

Provide a table mentioning various microbes, fermentation media and methods used, HoPS produced and their applications.

Line 309: HePS, composed of several carbohydrate moietie…lactose is not only the substrate for HePS. Make this statement more general and discuss reference ..Bioprospecting of exopolysaccharide from marine Sphingobium yanoikuyae BBL01: production, characterization, and metal chelation activity

Section 5.2: Author are suggested to prepare a table showing processing conditions and effect on EPS properties.

Author Response

This article discuss about variability of bacterial homopolysaccharide production and properties during food processing. Manuscript needs following changes.

Comments:

Line 40: One particularity of bacterial HoPS is that they can be produced at high level…not all the microbes able to produce that hight amount of EPS. Mention microbial examples able to produce high content of EPS.

A mention to bacterial species able to produce high level of HoPS was added.

Line 42: Many studies investigated EPS bioactivites, and it is now recognized…is mentioned activities shown by all EPS? You can discuss specific examples of EPS and their properties.

We added mentions about specific EPS and their properties, with relevant references (l. 56-61 in the revised manuscript).

In introduction part need to provide difference in hompolysaccharides in relation to composition, bioactivity, synthesis methods etc. What are the special properties of HoPS that make them different from other EPS. What is the importance of this article?

We added elements relevant to the specific interest of HoPS according to the reviewer suggestion. (l. 68-74 in the revised manuscript).

Provide a table mentioning various microbes, fermentation media and methods used, HoPS produced and their applications.

Table 2 was modified according to the reviewer suggestion.

Line 309: HePS, composed of several carbohydrate moietie…lactose is not only the substrate for HePS. Make this statement more general and discuss reference ..Bioprospecting of exopolysaccharide from marine Sphingobium yanoikuyae BBL01: production, characterization, and metal chelation activity

We corrected the sentence to be more general and added references to illustrate the production, the structure and the utility of HePS in dairy products (l. 339-343 in the revised manuscript).

Section 5.2: Author are suggested to prepare a table showing processing conditions and effect on EPS properties.

We added an additional table according to the reviewer suggestion (Table 3).

Reviewer 2 Report

The review is devoted to homopolysaccharide production and their properties, including when used in the food industry. The literary review includes 78 references. The list of keywords includes rheology of which, to my regret, little attention is paid in the review, and this is one of the key parameters when using such polymers to obtain products with the required properties. I would like to expand this part. As an example of work on the rheology of aqueous dispersions of bacterial cellulose, see - https://doi.org/10.3390/pr8040423. Small notes include: Line 35. The untrained reader may get the wrong conclusion about the water solubility of HoPS. For example, cellulose is not water soluble. Better to write that some polymers are water-soluble. Lines 71-73. Check "dextran 71 (between 9.0 × 106 and 5.0 × 108 Da)" and "The size of dextrans varies between 104 Da for Acetobacter tropicalis to 109 Da for Oenococcus kitaharae".

Author Response

The review is devoted to homopolysaccharide production and their properties, including when used in the food industry. The literary review includes 78 references.

The list of keywords includes rheology of which, to my regret, little attention is paid in the review, and this is one of the key parameters when using such polymers to obtain products with the required properties. I would like to expand this part. As an example of work on the rheology of aqueous dispersions of bacterial cellulose, see - https://doi.org/10.3390/pr8040423.

Thank you for these comments. We added a paragraph on rheological behaviour as suggested by the reviewer, in order to describe the relation between the structure of HoPS described in this review and the viscosity parameters that are explored in section 5.3 (l. 402-414 in the revised manuscript).

Small notes include: Line 35. The untrained reader may get the wrong conclusion about the water solubility of HoPS. For example, cellulose is not water soluble. Better to write that some polymers are water-soluble.

Correction has been made according to the reviewer suggestion.

Lines 71-73. Check "dextran 71 (between 9.0 × 106 and 5.0 × 108 Da)" and "The size of dextrans varies between 104 Da for Acetobacter tropicalis to 109 Da for Oenococcus kitaharae".

Correction has been made.

Reviewer 3 Report

biology-1513734-peer-review-v1

Variability of bacterial homopolysaccharide production and 2 properties during food processing

by Marion Nabot et al.

This review summarizes recent developments in the field of polysaccharides produced by microorganisms. It is sound from my point of view and some minor improvements are necessary.

L90/91: small caps should be used for D

L96-104: species names in italics

L139: Frutan or Fructan?

L178-L184: Maybe I did not get the point… the enzyme is discussed here, while resulting polysaccharides are topics in the other paragraphs

L408: “before being able” or similar

Author Response

This review summarizes recent developments in the field of polysaccharides produced by microorganisms. It is sound from my point of view and some minor improvements are necessary.

Thank you for your comments.

L90/91: small caps should be used for D

Done

L96-104: species names in italics

We verified the text and applied changes (l. 118-121 in the revised manuscript).

L139: Frutan or Fructan?

Corrected

L178-L184: Maybe I did not get the point… the enzyme is discussed here, while resulting polysaccharides are topics in the other paragraphs

An introductive sentence was added to ease the connexion of this paragraph with the others. (l. 200-201 in the revised manuscript).

L408: “before being able” or similar

Done

Reviewer 4 Report

I’m afraid that I found this review very difficult to read for two reasons. One, the sentences were often missing a word or two I was expecting and two, the reader was not really given an in depth enough recounting of the results in each of the studies mentioned. I was left with the feeling that if I wanted to learn anything I would have to read the original paper.

The title of the article seems a bit misleading.

Minor points:

Around line 70 the authors say molecular weights range between 104 to 106 but then give example of 107 and 108!

I would think the molecular weights in table 1 should be given a range or not be quoted to such high accuracy.

Mutan is not an α(1→6)Glc polymer.

Author Response

I’m afraid that I found this review very difficult to read for two reasons. One, the sentences were often missing a word or two I was expecting and two, the reader was not really given an in depth enough recounting of the results in each of the studies mentioned. I was left with the feeling that if I wanted to learn anything I would have to read the original paper. The title of the article seems a bit misleading.

This review intends to address the diversity of HoPS in structure, size and origin in relation with their interest for food application. The variability of HoPS nature was mainly investigated in model systems whereas the technological benefit of HoPS in foods is under the dependence of multiple in situ environmental parameters highlighted in this article. To the best of our knowledge, papers specific of regulation of HoPS production in foods were reported here and very few detailed HoPS composition or structure specificities. That is why this review aims to explore the opportunities provided through the control of HoPS production for lactic fermented food products optimisation.

The manuscript was corrected for English language.

Minor points:

Around line 70 the authors say molecular weights range between 104 to 106 but then give example of 107 and 108!

The molecular weights were corrected.

I would think the molecular weights in table 1 should be given a range or not be quoted to such high accuracy.

We respectfully remark that data in table 1 were reproduced here according to the original articles where accurate number were given to characterize the studied HoPS for molecular weights, however numbers were rounded off according to the reviewer comment.

Mutan is not an α(1→6)Glc polymer.

Correction has been made.

Round 2

Reviewer 1 Report

Author has revised the manuscript according to reviewers comments and can be accepted as it is.